# A Single-Port, Multiple-Access, Custom-Made Device Used in Laparoscopically Assisted Cryptorchidectomy in Standing Horses—A Preliminary Study

**DOI:** 10.3390/ani14071091

**Published:** 2024-04-03

**Authors:** Rafaela das Mercês Silva, Luiz Henrique Vilela Araújo, Thiago da Silva Cardoso, Stephany Lorrane Ishida Franco, Heytor Jales Gurgel, Pedro Henrique Lira Cerqueira, Lucas Santos Carvalho, Luis Gustavo e Silva Novais, José Leandro da Silva Gonçalves, Loise Araújo de Sousa, Rodrigo dos Santos Albuquerque, Marcos Duarte Dutra, Tatiane Teles Albernaz Ferreira, José Alcides da Silveira, Marco Augusto Machado Silva, Francisco Décio de Oliveira Monteiro, Pedro Paulo Maia Teixeira

**Affiliations:** 1Institute of Veterinary Medicine, Pará Federal University, Belém 68740-970, Brazil; rafaela.veterinaria@gmail.com (R.d.M.S.); henriqueevilela@gmail.com (L.H.V.A.); thiagodacardoso09@gmail.com (T.d.S.C.); stephanyyfranco@gmail.com (S.L.I.F.); jalesgurgel@gmail.com (H.J.G.); pedroh_lira@hotmail.com (P.H.L.C.); lucasfilhozaidan477@gmail.com (L.S.C.); gustavo.novais27@icloud.com (L.G.e.S.N.); leandrosilvag08@gmail.com (J.L.d.S.G.); loisesousa.a@gmail.com (L.A.d.S.); rdsa20@gmail.com (R.d.S.A.); duarte.marcos@gmail.com (M.D.D.); tatyalbernaz@ufpa.br (T.T.A.F.); jalcides2@gmail.com (J.A.d.S.); ppaulomt@ufpa.br (P.P.M.T.); 2Veterinay and Animal Sciense School, Federal University of Goiás (UFG), Goiânia 74690-900, Brazil; silvamam@gmail.com; 3Campus Araguatins of the Federal Institute of Education, Science and Technology of Tocantins (IFTO), Araguatins 77950-000, Brazil

**Keywords:** equine, minimally invasive surgery, multiport laparoscopic surgery, laparoscopic surgery, standing laparoscopy

## Abstract

**Simple Summary:**

This article evaluates a low-cost, multiport device for single access to the abdominal cavity of cryptorchid horses in the standing position. The device evaluation was initially carried out in five cadavers of bovine fetuses (*n* = 5) and later in four cryptorchid horses during routine hospital care. This study concludes that the device was efficient, allowing a wide exploration of the abdominal cavity, especially the inguinal region. Laparoscopically assisted cryptorchidectomy with the new device was feasible in two live animals and allowed the determination of the conditions of the retained testicles in the others, demonstrating the previous efficiency of the device in these cases.

**Abstract:**

This study evaluates a new multiport device with single access to the abdominal cavity produced with routine hospital supplies that could be applied to laparoscopically assisted cryptorchidectomy in standing horses. Initially, the new device was evaluated on five cadavers of bovine fetuses (*n* = 5), placed assisted in a minilaparotomy performed in the flank region. Subsequently, the device was evaluated in four cryptorchid horses treated during the hospital routine. During the evaluation of the new device, the possibilities of exploring the abdominal cavity, inspection, and intra-abdominal manipulation with two Babcock forceps were verified. The possibilities were described, and surgical time data were recorded and analyzed using descriptive statistics. In the cadavers, a wide exploration of the abdominal cavity was possible, with a laparoscopic inspection through the right paralumbar fossa and manipulation of intra-abdominal structures with Babcock forceps inserted by the new device. In cryptorchid horses, laparoscopically assisted cryptorchidectomy with a new device was feasible in two patients, and in the others, it allowed the diagnosis of adhesions and ectopic locations in the inguinal region of testicles retained in the cavity. Therefore, the new device was efficient in exploring the inguinal region of cryptorchid horses in the standing position. The present study is preliminary and can support future studies that aim to improve the developed prototype.

## 1. Introduction

Laparoscopic cryptorchidectomy is one of the most commonly performed procedures in the field of equine genitourinary surgery, despite limitations related to the skill of the surgeon and the use of specialized equipment [1,2,3]. Cryptorchidectomy assisted with laparoscopy can allow better intra-abdominal inspection, facilitating access and manipulation of the testicle retained in the cavity [1,4,5].

In recent decades, there have been many advances related to laparoscopic cryptorchidectomy in horses, the main benefits of which are the absence of general anesthesia and better visualization of the cryptorchid [1,6,7]. Standing horse laparoscopy is the best option to remove ectopic testicles, but complications can occur when the devices are not used during access to the abdominal cavity [8,9].

Related advances to laparoscopic surgery depend on improvements and related innovations to the technique and equipment [1,9,10]. The most recent studies demonstrate single access through an umbilical port for the removal of cryptorchid testicles and the use of devices such as wound retractors and resorbable self-locking loops [7,11,12].

No report was seen by the authors on the use of a low-cost, multiport device for laparoscopic assisted cryptorchidectomy in horses. The objective of this study was to develop and evaluate a new low-cost, multiport device to perform laparoscopically assisted cryptorchidectomy in horses. Our hypothesis is that it would be possible to develop a new low-cost, multiport device to perform laparoscopically assisted cryptorchidectomy in horses.

## 2. Materials and Methods

### 2.1. Study Site and Multiport Device Evaluation Steps

This study corresponds to the development of a new low-cost, multiport, self-innovated device for performing laparoscopically assisted cryptorchidectomy in standing horses. the first stage of evaluation of the multiport device was carried out on cadavers of bovine fetuses from cows in the last third of pregnancy slaughtered in a local slaughterhouse that serves the requirements of the Brazilian health inspection.

The experiment was carried out at the Institute of Veterinary Medicine (IMV) of the Federal University of Pará (UFPA), where the low-cost, multiport device for laparoscopically assisted cryptorchidectomy was developed in standing horses.

The first stage of evaluation of the multiport device was carried out on five cadavers of bovine fetuses weighing 30 to 40 kg (AC, *n* = 5), in which the feasibility of laparoscopy through the right flank was evaluated using the developed multiport device. The second stage of evaluation of the multiport device was carried out in live animals: four cryptorchid horses treated at the IMV/UFPA Veterinary Hospital, in vivo evaluation (AV, *n* = 4).

### 2.2. Preparation of the New Single-Port, Multiple-Access, Custom-Made Device

The single-port, multiple-access, self-innovated device was prepared before the start of the surgery with a cone-shaped sterile polypropylene structure measuring 9.0 cm in height, 9.5 cm in diameter at its base, and 5.5 cm in diameter at its apex (Figure 1). A pair of sterile powder-free surgical gloves (size 8.5; Medix Brasil, São Paulo, Brazil) was used to cover this polypropylene structure so that the fingers of the gloves covered the base; that is, the polypropylene structure was placed inside of the gloves, overlapping with their 9.5 cm base facing the fingers of the gloves, forming the multiple-access of the self-innovated device. The proximal portion of the device, where the ring finger of the glove was used as an access portal for the 5 mm Babcock laparoscopic forceps, and the middle finger of the glove was used as an access portal for the 10 mm Babcock laparoscopic forceps (Figure 1).

A surgical glove was placed over the polypropylene structure from the base to the apex, surrounding it. A ring with an uncuffed endotracheal probe (size 2.5; Solidor, São Bernardo do Campos, São Paulo, Brazil) was mounted in the apex region. The folded edge of the glove cuff was wrapped around this ring and fixed through a separate single stitch using a nylon thread (2-0; Shalon Medical, Goiânia, Goiás, Brazil) (Figure 1).

The tips of the middle and ring fingers of the gloves were cut with sterile scissors after the distal ring of the device was inserted into the abdominal cavity. From the remaining part of the fingers, two pieces were cut to form two elastic bands. Next, two laparoscopic Babcock forceps (5 mm and 10 mm; Karl endoscopeoskope, São Paulo, São Paulo, Brazil) were inserted into each finger of the glove that was cut, ring and middle. Each of the prepared elastic bands was used to hermetically connect the cut fingers of the glove to the Babcock clamps.

### 2.3. Instruments and Equipment Used in this Study

The evaluation stages of the multiport device in bovine fetal cadavers and in cryptorchid horses took into account all the surgical principles applicable to laparoscopy and conventional open surgery, and the necessary equipment and instruments were used to perform the techniques. We used a 10 mm laparoscope (Karl Storz—hopkins, São Paulo, São Paulo, Brazil), 10 mm or 5 mm Babcock forceps (Karl Storz—endoskope, São Paulo, São Paulo, Brazil), 5 mm laparoscopic scissors (Karl Storz—Clickline, São Paulo, São Paulo, Brazil), a set of gas insufflator/light source/monitor (Karl Storz—endo-arthroflator-vet, São Paulo, São Paulo, Brazil), and basic surgical instruments for conventional surgery (ABC Surgical Instruments, São Paulo, São Paulo, Brazil).

### 2.4. Evaluation of a Customized Single-Port, Multiple-Access Device in Bovine Fetal Cadavers

The evaluation of the multiple-access device in bovine fetal cadavers was to verify the feasibility of using this device and introducing instruments into the abdominal cavity. Bovine fetuses were used because they are constantly used in our routine investigations and discoveries of videosurgical techniques. The cadavers were placed in the left lateral decubitus position and subjected to laparoscopy using the multiport device on the right flank. The first laparoscopic access port was in the paralumbar fossa, caudal to the last rib, and ventral to the transverse processes of the third and fourth lumbar vertebrae, with direct insertion of the trocar carefully and slowly, initially with an angle of 45° in relation to the abdominal wall and subsequently 90° (Figure 2A). A skin incision of approximately 8 to 10 mm was made with a scalpel for transmural insertion of the 10 mm trocar with an insufflation valve for insufflation into the abdominal cavity. The correct position of the trocar in the peritoneal cavity was verified by intra-abdominal inspection using the laparoscope that was introduced into the trocar cannula. The pneumoperitoneum was induced with carbon dioxide (CO_2_) injected through the inflation valve. Laparoscopic inspection of the abdominal cavity was performed with the abdomen distended by CO_2_, expanding the visual field.

An intra-abdominal inspection was performed to identify the placement of the device on the right flank of the cadavers. A skin incision measuring approximately 3 cm was made below the laparoscopic port, and subsequently, a dissection of the subcutaneous and muscular layers was performed to access the abdominal cavity through this minilaparotomy (Figure 2A,B).

Through a minilaparotomy, with the aid of Collin forceps, the distal ring of the device was introduced into the abdominal cavity. The laparoscopic forceps were placed in the multiple-access portion through the fingers of the glove. Elastic bands were placed on these fingers after the clamps to prevent CO_2_ leakage. The exploration of the abdominal cavity was performed using the new self-innovated device (Figure 2C). After intra-abdominal exploration with a new device, it was removed and the pneumoperitoneum was undone, followed by myorrhaphy with a separate X suture using a nylon-0 thread. The skin suture was performed in a simple isolated pattern using the nylon-0 thread.

### 2.5. Evaluation of the New Device in Cryptorchid Horses

The evaluation of the new single-port, multiple-access, self-innovated device in live horses was carried out. Four cryptorchid horses over three years of age of the Manga-larga, Lusitano, quarter horse, and crossbreed breeds presented at the veterinary hospital for routine laparoscopic cryptorchidectomy were the subjects of this study.

The clinical examination was carried out on all patients according to routine hospital care, including a preoperative hematologic evaluation based on complete blood count tests. After 12 h fasting, the patient was restrained standing.

The tranquilization and sedation of each patient were induced in the containment trunk with an intravenous injection of 1% acepromazine (Acepran^®^, Vetnil, Louveira, São Paulo, Brazil) associated with 1% detomidine (Detomidin^®^, Syntec, Santana de Parnaíba, São Paulo, Brazil) at doses of 0.03 mg/kg and 0.02 mg/kg, respectively. By continuous infusion, at a dosage of 20 microdrops per minute, a lactated Ringer’s solution (Ringer com lactate, JPFarma, Ribeirão Preto, São Paulo, Brazil) containing 1% detomidine (Detomidin^®^, Syntec, Santana de Parnaíba, São Paulo, Brazil) and 1% butorphanol (Butorfin^®^, Vetnil, Louveira, São Paulo, Brazil) was administered at doses of 0.02 mg/kg and 0.013 mg/kg, respectively.

A thoracolumbar paravertebral anesthetic block was performed to provide analgesia at the surgical site (Figure 3C). An antisepsis from the paravertebral region to the dorsal region of the flank was initially performed with clipping and an antiseptic based on 2% chlorhexidine digluconate and alcohol (Riohex^®^, Rioquímica, São José do Rio Preto, São Paulo, Brazil) (Figure 3A). Blocks of the dorsal branches of the spinal nerves T18, L1, L2, L3, and L4, the proximal paravertebral block, were performed with the subcutaneous infiltrative deposition of 5 mL of 2% lidocaine (Lidovet^®^, Bravet, Rio de Janeiro, Brazil) using a spinal needle coupled to a 20 mL disposable syringe that was inserted immediately cranially into the transverse processes of the T18, L1, L2, and L3 vertebrae (Figure 3B). For blockages of the ventral branches of the nerves, the distal paravertebral block, the catheter progressed through the intertransverse ligament and subsequent infiltrative deposition of 5 mL of local anesthetic (Figure 3B).

The surgical site was prepared after paravertebral anesthesia with the horse restrained in the containment trunk under sedative effect, where antisepsis was performed by placing field cloths in the paralumbar fossa and the ipsilateral flank of the testicle retained in the abdominal cavity (Figure 4).

A laparoscopic port was established in the center of the paralumbar fossa, caudal to the last rib and ventral to the transverse processes of the lumbar vertebrae, midway between the last rib and the tuber coxae (Figure 5A). A vertical incision of approximately 6 mm was made through the skin to introduce a 10 mm laparoscopic cannula with a blunt tip obturator through the incision into the abdominal cavity carefully, with alternating rotation movements, clockwise and counterclockwise. Subsequently, the obturator was removed, and a laparoscope was inserted through the cannula for intra-abdominal visualization with the coupling of the CO_2_ insufflation hose, through which a pressure pneumoperitoneum of 8 mmHg was established with a flow speed of 5 L/min (Figure 5B).

Intra-abdominal laparoscopic inspection was performed immediately by introducing the rigid endoscope through the laparoscopic port. This allowed an assisted minilaparotomy to be performed on the flank, below the laparoscopic portal (Figure 5A), through which the new single-port, multiple-access, self-innovated device was placed with the aid of Collin forceps (Figure 5B), inserting the distal ring of the device inside the abdominal cavity (Figure 5C).

Through the middle and ring fingers of the inserted device, Babcock-type laparoscopic forceps were inserted for intra-abdominal manipulation of internal structures associated with or adjacent to the cryptorchid testicles (Figure 5A,C). The hermetic sealing of the previously cut glove fingers, which contain the laparoscopic forceps, was performed with the elastic bands obtained during the cuts of the glove fingers.

Cryptorchid testicles and associated and adjacent structures were inspected laparoscopically, removed from the inguinal ring, and moved to the minilaparotomy with the help of Babcock forceps. The minilaparotomy was expanded after the multiport device to facilitate extra-abdominal exposure of the cryptorchid testicle captured by the forceps. The spermatic cord was transfixed with a double ligature outside the abdominal cavity, using a 2-0 polyglycolic acid surgical thread, and then sectioned and extracted from the abdominal cavity.

An intra-abdominal lavage with lactated Ringer solution (500 mL) mixed with 2% lidocaine (2 mL) was performed after cryptorchidectomy.

The two surgical incisions, the laparoscopic portal and the minilaparotomy, were sutured with a 2-0 polyglycolic acid thread. Minilaparotomy myorrhaphy was performed with sutures in an ‘X’ pattern and subcutaneous reduction with sutures in a simple continuous pattern. Dermorrhafia of both incisions was performed with intradermal stitches.

### 2.6. Postoperative of Horses Undergoing Cryptorchidectomy

Postoperatively, the horses were subjected to a medication protocol based on phenylbutazone (Fenilbutazona OF^®^, Ourofino, Minas Gerais, Brazil), at a dose of 4.4 mg/kg, intravenously, once a day for five days; omeprazole (Equiprazol^®^, Vetnil, São Paulo, São Paulo, Brazil), at a dose of 4 mg/kg, orally, once a day for five days; and antibiotics associated with benzathine, benzylpenicillin, procaine, and potassium with dihydrostreptomycin and streptomycin (Pencivet^®^ Plus PPU, MSD Saúde Animal, São Paulo, São Paulo, Brazil), at a dose of 5 mg/kg, intramuscularly, once a day for five days. The patient was kept in a stall for five days, and his physiological parameters (HR—heart rate, RR—respiratory rate, and rectal temperature) were monitored. Surgical wounds were dressed with chlorhexidine and ointment until the patient was discharged, and wound treatment continued until the patient’s complete recovery.

### 2.7. Statistics

Data were subjected to descriptive statistics, recording possible failures and complications, and measuring operative time. All tests were performed using the BioEstat 5.3 statistical package.

In the evaluation using a cadaver model, the time of the following operative steps was measured in minutes: placement of the laparoscopic port, placement of the multiport device, abdominal exploration, laparorrhaphy, and total procedure time. When evaluating live animals, only the total cryptorchidectomy time was measured in minutes.

Time data were presented as mean ± standard deviation, and complications were presented in absolute and relative frequency.

## 3. Results

The initial performance of the experiment was preceded by the assembly of the multiport device (Figure 1); this assembly was quick and simple, not exceeding five minutes to complete the assembly of the multiport device, as can be seen in Figure 1. 

### 3.1. Evaluation of the New Single-Port, Multiple-Access, Self-Innovated Device in Bovine Fetal Cadavers

The initial evaluation of the device in bovine fetal cadavers was performed in five experimental models (5/5, 100%); it was feasible to perform the assisted laparotomy technique using the new multiport device and a laparoscopic portal (Figure 2). The technique allowed us to explore the abdominal cavity of bovine fetuses, including the inguinal region.

The multiport device allowed intra-abdominal manipulation of internal structures and organs with two Babcock forceps inserted into the ring and middle fingers of the glove. The airtight seal of these fingers on the forceps secured with elastic bands was efficient and allowed the pressure of the pneumoperitoneum to be maintained. The pressure was checked in the CO_2_ insufflator, where both the CO_2_ injection speed and the intra-abdominal pressure were monitored.

Table 1 below presents the times.

The flexible portion of the multiport device, the surgical glove on the laparoscopic forceps, ruptured twice during the procedure in the same experimental model (1/5, 20%).

### 3.2. Evaluation of the New Single-Port, Multiple-Access, Self-Innovated Device in Cryptorchid Horses

Laparoscopically assisted cryptorchidectomy with a new device was feasible in three horses (3/4, 75%), as in one animal it was not possible to place the new device due to the greater thickness of the abdominal wall (1/4, 25%).

Minilaparotomy in the flank region allowed placement of the distal ring of the multiport device in the three patients, ensuring intra-abdominal manipulation of the cryptorchid testicles with the two clamps inserted by the new device. This manipulation, with a laparoscopic apparatus, allowed the removal of cryptorchid testicles in two patients (2/4, 50%).

In two patients, it was not possible to exteriorize the cryptorchid testicle with the new device. One patient had a very thick abdominal wall, making placement of the device difficult. Another patient had the testicle strongly adhered to the abdominal tissues, making it impossible to pull it out of the cavity. Therefore, the device was removed, and the minilaparotomy was expanded to perform adhesionolysis with one hand (Figure 6A) and subsequent exteriorization of the testicle (Figure 6C). The testicle retained in the abdominal cavity was externalized by manual traction of the pampiniform plexus (Figure 6A–C).

Double ligation of the sperm cord was effective in containing internal bleeding. Laparorrhaphy in three planes in minilaparotomy (muscular/peritoneum, subcutaneous, and skin suture with intradermal suture) was also efficient in closing the surgical incision of the abdominal wall (Figure 7A). Only an intradermal suture was effective in closing the surgical incision of the laparoscopic portal (Figure 7A). The surgical wounds were approximately 1 cm in size at the laparoscopic port access site and 2.5 cm at the abdominal access site of the multiport device (Figure 6D and Figure 7A).

Laparoscopically assisted cryptorchidectomy using the new single-port, multiple-access, self-innovated device was feasible and allowed the removal of cryptorchid testicles from two patients (Figure 7B,C), as well as the identification of the status of the cryptorchid testicles from the other patients. The average surgical time was 204 ± 9.9 min, with a minimum time of 190 min and a maximum time of 212 min. 

Postoperative treatment based on phenylbutazone, antibiotics comprising penicillin and streptomycin, hospital admission, and daily dressings of the surgical wound helped the patient recover from surgery in ten days.

## 4. Discussion

The new single-port, multiple-access, self-innovated device, assembled with everyday hospital supplies, is a low-cost discovery designed to perform cryptorchidectomy on a standing horse. The inputs used can be improved and even adapted to each reality since the assembly of the device is simple. We believe that it is an important prototype; although this study is preliminary, this device can be improved and adapted to single-site laparoendoscopic (or LESS) approaches. Advances related to cryptorchidectomy in standing horses depend on improvements and discoveries using equipment such as this [1,9,10].

The device was initially evaluated in bovine fetal cadavers as it had never been used or tested on live animals. Preliminary testing of equipment and surgical techniques on cadavers is essential to improve procedures and prototypes, as well as to obtain technical skills and equipment use [13,14,15,16].

Placing the device on the flank of bovine fetal cadavers was possible in all models because they have a thinner abdominal wall compared to an adult animal with more developed muscles [17,18]. Initially, the laparoscopic portal was established in the paralumbar fossa of the cadavers and provided a wide inspection of the abdominal cavity, supporting the previous conclusion that laparoscopic exploration through the flank of the standing horse would be possible. The laparoscopic approach through the paralumbar fossa allowed assisted performance of a minilaparotomy and placement of the new device, facilitating these procedures by providing an intra-abdominal view and providing greater safety to surgeons due to the increased viewing angle [13,19].

Laparoscopically assisted cryptorchidectomy in standing horses has gained popularity as a therapeutic and diagnostic tool, as it provides excellent intra-abdominal visualization, easy localization of the undescended testicle, reduced complications, and rapid recovery [8,20]. In this way, developing equipment that is easy to assemble and low-cost can improve and perfect intra-abdominal exploration of the retained testicle, contributing to the improvement of the technique and its increased use and seeking a rapid recovery of the patient with fewer postoperative complications.

The surgical scar was reduced as it was a minimally invasive procedure, and the small size of the removed testicle was another important factor in determining the size of this scar.

The new device evaluated in live animals proved to be efficient for cryptorchidectomy of testicles retained in the inguinal region in cases not complicated by adhesions [6,9]. Laparoscopy with a new device allowed a broad intra-abdominal inspection and tissue manipulation in the inguinal region of standing horses, a place where cryptorchid testicles are commonly retained, allowing easy identification of the gonadal and an immediate diagnosis of these testicles [1,7].

The development of safer equipment to improve abdominal access in standing horses may result in procedures with fewer complications, considering that complications such as organ perforation, difficulty crossing the abdominal wall, and peritoneal displacement of the peritoneum can occur [5]. In the prototype developed, the flexible part, made up of gloves, made it difficult at times to handle the forceps in the abdominal cavity due to its great flexibility and length, therefore requiring adaptation since unsafe handling can lead to complications in the transoperative or post-operative.

The flexibility of the device at times also made it difficult to place it to access the abdominal cavity, influencing the total surgery time since, when the flexible portion broke, it was necessary to assemble a new device and restart its placement through the minilaparotomy. In this case, it is suggested that the flexible portion of the new device be produced with a more consistent, resistant, versatile, durable, and low-cost material, such as synthetic or semi-synthetic plastic, or even with a material produced by latex [21,22].

Surgical time was longer in live animals due to the complexity of the cases and the greater thickness of the abdominal wall. Among the causes that can influence the surgical time of laparoscopically assisted cryptorchidectomy in standing horses, we can mention the location of the cryptorchid testicle (inguinal vs. abdominal), temperament, size, and age of the horse, the surgeon’s technical skill, and inadequate instruments and equipment [5,6,8].

The new self-innovated device was efficient during intra-abdominal exploration of cryptorchid horses. Performing laparoscopically assisted cryptorchidectomy with the device was feasible, with few failures and complications. This study is preliminary and has limitations, such as a low number of cryptorchidectomies.

The results of this study provide the first information about the development of a new device for a laparoscopic approach to cryptorchid horses in the standing position, serving as a basis for more standardized studies that investigate this topic. 

## 5. Conclusions

The handcrafted multiport device developed provided access to the abdominal cavity of horses undergoing laparoscopically assisted cryptorchidectomy, allowing intra-abdominal manipulation and removal of the cryptorchid testicles located in the inguinal region.

Laparoscopically assisted cryptorchidectomy using the new single-port, multiple-access, self-innovated device was feasible and provided an examination of the ipsilateral half of the abdominal cavity.

The evaluation of the multiport device in bovine fetal cadavers was important to confirm its effectiveness in guaranteeing access to the abdominal cavity.

Further studies are needed to develop alternatives capable of improving the multiport device, making it more resistant and adaptable to different thicknesses of equine abdominal walls.

## Figures and Tables

**Figure 1 animals-14-01091-f001:**
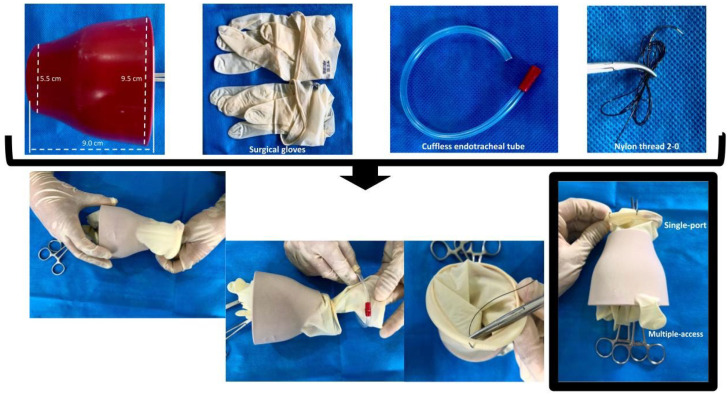
Preparation of the single-port of the self-innovated device.

**Figure 2 animals-14-01091-f002:**
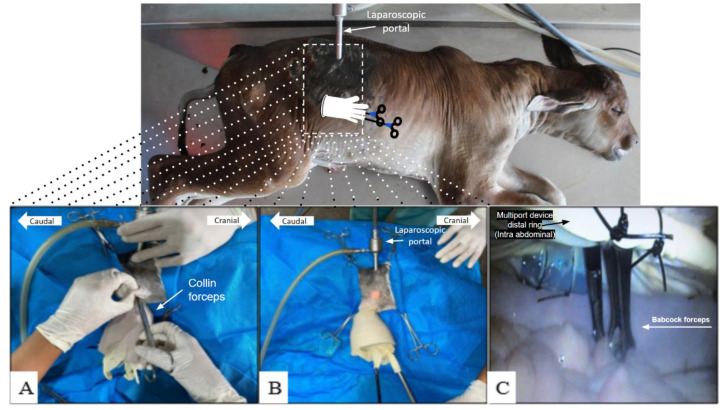
Intra-abdominal access using the single-port of the self-innovated device in bovine cadavers: (**A**) placement of the single-port of the self-innovated device in the minilaparotomy with the aid of Collin forceps; intra-abdominal insertion of the distal ring of the single-port of the self-innovated device; (**B**) placement of the single-port of the self-innovated device; distal ring introduced into the abdominal cavity through minilaparotomy; laparoscopic forceps placed and hermetically sealed in the multiple-access of the device, ring, and middle fingers of the glove. (**C**) intra-abdominal exploration using the single-port of the self-innovated device; view of tissue manipulation with laparoscopic forceps in the abdominal cavity.

**Figure 3 animals-14-01091-f003:**
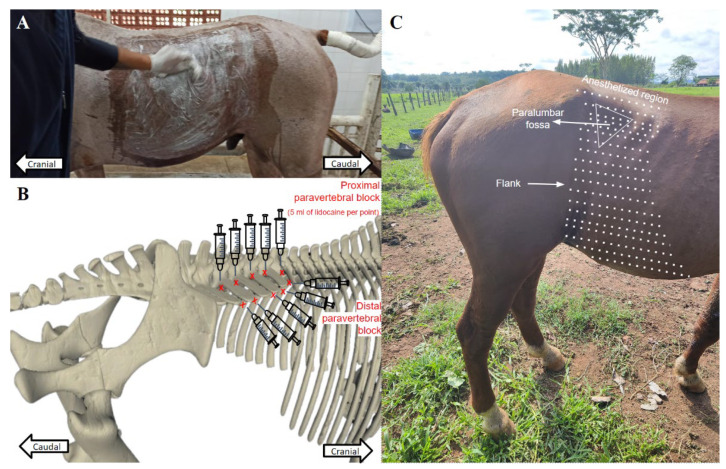
Paravertebral anesthesia performed on horses undergoing video-assisted cryptorchidectomy: (**A**) wide antisepsis of the abdominal quadrant, paralumbar fossa, flank, and paravertebral region; (**B**) schematic demonstration of dorsal and distal paravertebral anesthetic blocks in horses undergoing video-assisted cryptorchidectomy; (**C**) region anesthetized by dorsal and distal paravertebral blocks.

**Figure 4 animals-14-01091-f004:**
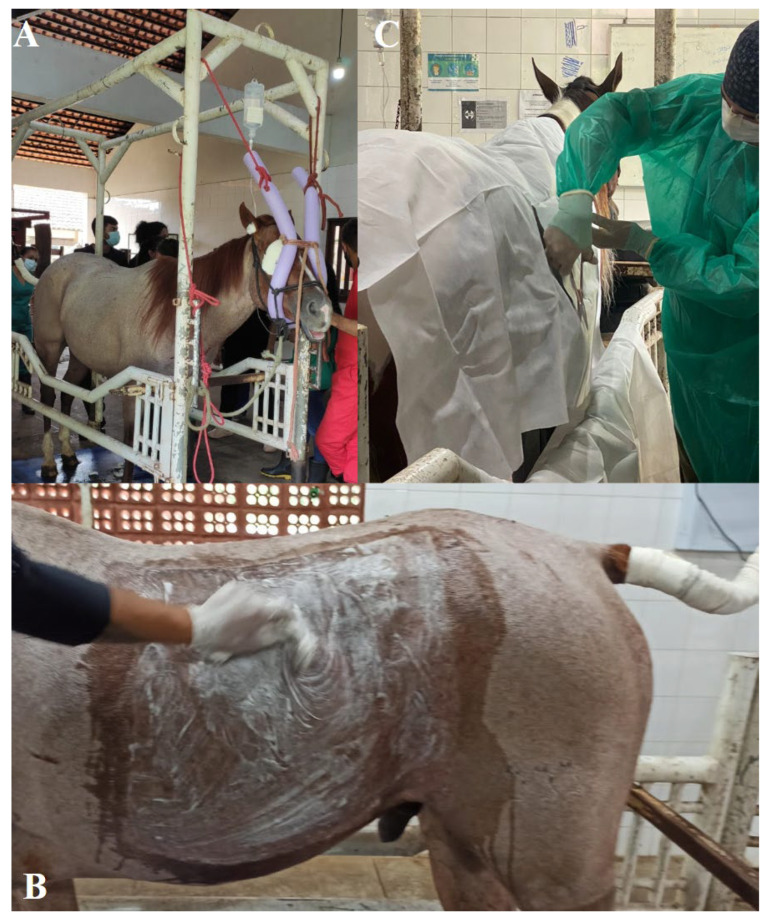
Restraint and antiseptic preparation of the patient while standing: (**A**) physical restraint in stocks suitable for the equine species; (**B**) wide antiseptic preparation of the abdominal quadrant ipsilateral to the cryptorchid testicle; (**C**) placement of drapes and antiseptic preparation of the surgical site.

**Figure 5 animals-14-01091-f005:**
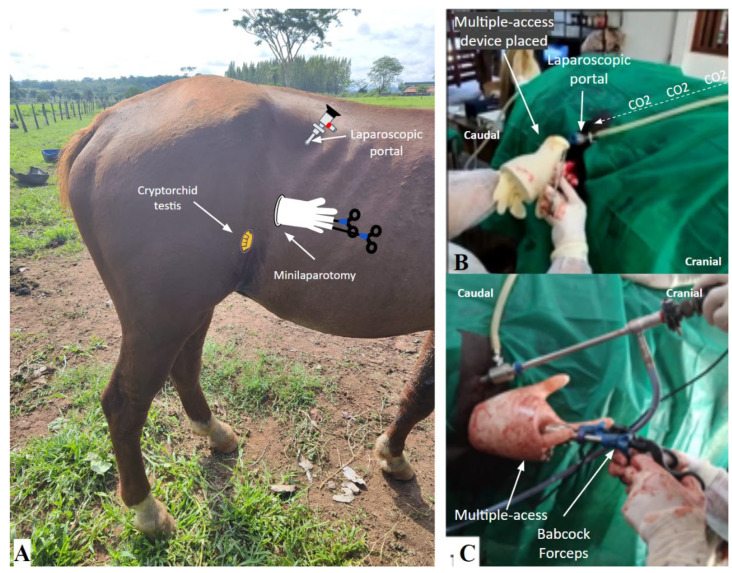
Laparoscopically assisted cryptorchidectomy using the new homemade single-port, multiple-access device in a standing horse: (**A**) demonstration of the location of the laparoscopic port, multiport device, and cryptorchid testis in the paralumbar fossa and flank of a standing horse; (**B**) demonstration of the surgical site with the placement of the multiport device with the aid of Collin forceps; (**C**) demonstration of the layout of the laparoscopic portal and multiport device in a procedure to explore the abdominal cavity of a standing horse.

**Figure 6 animals-14-01091-f006:**
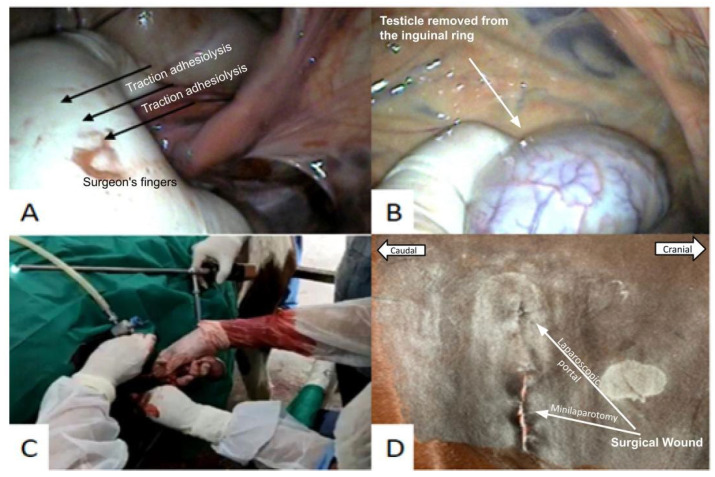
Laparoscopically assisted cryptorchidectomy of the testicle attached to the inguinal region: (**A**) adhesiolysis of the cryptorchid testicle of the inguinal region performed with the surgeon’s hand; (**B**) testicle outside the inguinal ring after traction with the surgeon’s hand; (**C**) double transfixation of the spermatic cord outside the abdominal cavity; (**D**) scars of surgical incisions made to establish the laparoscopic portal and place the new device (minilaparotomy).

**Figure 7 animals-14-01091-f007:**
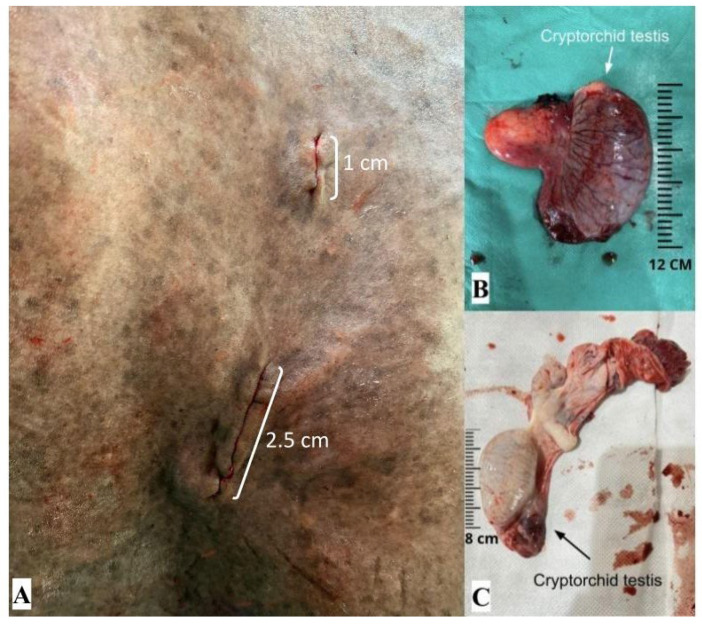
Final result of laparoscopically assisted cryptorchidectomy using the new single-port, multiple-access, self-innovated device; (**A**) approximate size of surgical wounds after laparoscopically assisted cryptorchidectomy; (**B**) cryptorchid testicle removed from a patient; (**C**) cryptorchid testicle removed from the other patient; cm: centimeter.

**Table 1 animals-14-01091-t001:** Average time in minutes of the surgical steps of laparoscopy using a new multiport, self-innovated device, tested on bovine fetal cadavers.

Laparoscopic Portal	Multiport Device Placement	Abdominal Exploration	Laparorrhaphy	Total Time
2 ± 1.2	8 ± 6	2.2 ± 2.1	7.4 ± 3.4	17.4 ± 8.2

## Data Availability

Data are contained within the article.

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
