# Peer review of "A Single-Port, Multiple-Access, Custom-Made Device Used in Laparoscopically Assisted Cryptorchidectomy in Standing Horses—A Preliminary Study"

_animals, 2024, doi:10.3390/ani14071091_

Round 1

Reviewer 1 Report

Comments and Suggestions for Authors

You should check the text because there are numerous spelling errors.

I believe the paper is not very original, especially aimed at cryptorchidism, when in this type of surgery the laparoscopic ports are always very small. The original thing is the cost of hand-made equipment.

It could be more interesting in ovariectomies, even if in my opinion we lose the mini-invasiveness character.

Author Response

Dear reviewer

Our team greatly appreciates your contributions to the manuscript, they were essential and enriching.
We hope we have addressed all your suggestions.
We remain at your disposal to meet other demands that you deem necessary for the manuscript to be published.

Best Regards,

Francisco Décio de O Monteiro
Author

Reviewer 2 Report

Comments and Suggestions for Authors

line 43 in abstract, statement that the prototype device will generate other discoveries related to the laparoscopic cryptorchidectomy technique in standing horses is very vague and not founded on much aspects of the report paper

lines 53 54, this is an assumption but I don't think your references demonstrates this statement, better should compare to another type of procedure, can you be more specific

line 56 sentence misses a verb

line 60 typo: space needed removal of the ectopic testicle, sentence should be shortened for clarity of reading,

line 62 vesselssit-upscan???

line 62-63: I don't think this statement is supported or established by the two references provided

line 64-65 spacing needed advances to; cryptorchidectomy laparoscopic; discoveries to; technique and

paragraph line 64 to 69 needs review of grammar and scientific writing (ex: seen vs found...)

line 71 sentence construction is inverted and spacing/punctuation not respected

line 83 spacing

line 86 and earlier quadrupedal position sounds very unusual, why not using standing position?

Is there a formal animal ethics committee application and formal approval with reference number for this experimental research?

line 88 + line 89 spacing

line 92 grammatical order incorrect

line 97 it is unclear whether the polypropylene structure is sterile or not. I have read the description on how to of build the homemade device but I can hardly visualise how this is put together. please use better figures/illustrations

line 110 spacing

paragraph line 120 to 125, a more detailed description with names and references of the manufacturers is advisable

line 126 grammatical structure inverted

line 127 to 129: I think I understand what you try to say but the sentence isn't clear

line 128 spacing(s)

line 130 spacing (2) and missing a "to"

line 132 space and transverse process of which lumbar vertebrae? can you decribe more precisely the direct insertion of the trocar in a careful manner, it is confusing which trocar you are talking about, how is a 10 mm trocar passing through a 5 mm skin incision?

line 136 spacing, as peritoneum tenting and retroperitoneal insufflation is one of the most common complications of laparoscopic surgery described in horses, how did you ascertain the correct position of this first trocar inside the peritoneal cavity prior to CO2 insufflation?

line 137 grammar and spacing

line 138 spacing - now talkiing about canula instead of trocar, sentence needs to be broken down in smaller sentences as there different actions performed

line 149 grammar and spacing

line 150 as this is the description of the preliminary study on cadavers, it is probably not necessary to repeat this here

line 152 divulsion describes a more brutal tearing, would you rather chose division/dissection or separation?

line 154 as the apex of the device is 5.5 cm and the skin incision for the mini laparotomy only 3 cm (line 151) how is the device fitted inside the abdominal wall?, skin is very inelastic and stretching to insert a larger instrument would cause undue trauma that can facilitate surgical site complications

line 156 maintaining the seal with fingers seems a very approximate technique, was there any objective measurement of the intraabdominal pressure performed?

line 158 inversion of words to describe the device

line 159 grammar and spacing

line 161 unconventional description of suture pattern, what suture material used for the myorrhaphy (single or mutliple layers?), by simple isolated do you single interrupted?

line 163 word inversion

paragraph line 164-167 grammatical construction to review

line 168 what do you mean by non-simultaneously?

line 171 do you mean restraining stocks or crush?

line 178 typo: words within brackets have been misplaced at the end of the sentence

line 183 replace trichotomy by clipping

line 184 suppress degerming and change alcoholic for alcohol

line 188 and 190 do you mean spinal needle instead of epidural catheter?

legend Fig 3C region anaesthetised by or thanks to instead of due to

line 121 stocks or crush not trunk

line 211 coxal tuberosity or tuber coxae instead of thigh tubercle, here again there is an undue disproportion betwwen the skin incision and the size of the laparoscopic instrumentation

line 249 dermorrhaphy - typo

line 251 post-operative... requires a word

paragraph line 252 - 260. to remain consistent drugs brand names and manufacturers should be reported, exact dosage /kg needs to be reported

what is the justification of not premedicating the horses with antiinflammatories and antimicrobials prior to surgery as the pharmakokinetic of theses drugs would require

line 273 by inputs do you mean components or parts?

line 277 word inversion homemade should come before device

line 291 as mentioned above, how was it ascertained that the intraabdominal pressure was maintained?

line 292 would you rather describe this as a laparoscopy assisted cryptorchidectomy than assisted laparoscopy?

line 300 word inversion required

line 302 I would definitely describe this technique as laparoscopically assisted cryptorchidectomy

line 320 what do you mean by extracavitary testicle?

line 334 how can the minilaparotomy incision of 2.5 cm accommodate a 5.5 cm diam apex of the homemade device unless the dermorrhaphy was intentionnally tightened very strongly

line 371 to 373, very confuse narrative with grammatical errors

line 378 do you mean the length of general anaesthesia instead of delay?

line 383 explain how equipment that is easy to assemble and of low-cost may improve the intraabdominal exploration, improve which technique and fewer complications. Your device is low cost but requires the initial use of the most costly equipment of laparoscopy: the laparoscope, one trocar, a processor, a light source, a monitor, a CO2 insufflator. Your technique saves 2 additional trocars and possibly the use of a dedicated laparoscopic electrocautery equipment (vessel sealing and dividing device of some sorts). On the other hand, the much larger incision necessary to insert the homemade device will increase the prevalence of surgical site complication/infection as it has been largely documented in other studies and if, in the case of cryptorchidectomy, the testicular pedicle is shorter for any given reason, you have to enlarge the incision to be able to insert the hand inside the abdomen, negating the advantages of minimal invasive surgery

line 390, this device is not modifying the potential for a broad exploration of the abdominal cavity as this is performed by the laparoscope inserted through the "classic" trocar inserted into the paralumbar fossa, you can't claim it enabled tissue manipulation in the inguinal area

line 393 change sperm structure for genital, gonadal or alike

line 395 this device using a direct minilaparotomy technique although laparoscopically controlled contradicts the conclusions of your reference (9)

line 401 grammar: in the trans- or postoperatively is incorrect grammatically

line 405-406: this sentence is not bringing any useful information to the discussion

line 409 I can't find the references to the discussion about latex or equipment improvement in the two papers cited (24)(25)

line 410 this discussion contradicts the results where the surgical times for cadavers is way shorter then live patients

paragraphs line 410 to 422 this section fluctuates between a vague very difficult to understand discussion and fragments of what would rather be in the introduction

the final portion of the discussion is extremely confuse and addressed concepts that are not well explained at all, the last paragraph in particular is full of grammatical errors and very imprecise ideas for future research

conclusion is a succession of repetition and ideas are commented in a sequence that is not following the working hypothesis order

considering the experimental and technicql protocol, I would strongly suggest to change the title for "Single-port multiple-access custom made device for laparoscopic assisted cryptorchidectomy in standing horses - a preliminary study"

Comments on the Quality of English Language

please refer to prior comments section, the overall quality of English is very average to poor and needs extensive editing

Author Response

(The authors gave the same response as above.)

Reviewer 3 Report

Comments and Suggestions for Authors

Your paper describes a single port technique for laparoscopy in horses with a homemade device. Unfortunately, the technique cannot be claimed as a "single port" technique as you still have to ports (camera and your device). Furthermore, the results do not seem suffieciently strong for publication in my opinion. 

Please find more comments on the attached document.

Comments on the Quality of English Language

I'm not an English native, but I see several paragraphs difficult to understand and errors in syntax (please see on the attached document).

Author Response

(The authors gave the same response as above.)

Reviewer 4 Report

Comments and Suggestions for Authors

I suggest that the authors include in their further studies to position their device as a valuable and useful model to perform construct, content, and face validity using the fetal bovine model.  Performing these experiments would undoubtedly provide veterinary surgeons with a valuable device to perform laparoscopic-assisted cryptorchid orchiectomy and other abdominal-related procedures in horses.

According to the audience focus of this work, it is not necessary to recall the importance of providing new models to perform training and acquisition of basic, advanced, and procedure-specific laparoscopic skills in any field of veterinary laparoscopic surgery.

Comments on the Quality of English Language

Author Response

(The authors gave the same response as above.)

Reviewer 5 Report

Comments and Suggestions for Authors

Reviewer comments for manuscript ID animals -2891311 entitled ‘New single-port multiple-access device homemade to laparoscopic cryptorchidectomy used in standing horses’

General Comments

Minimally invasive surgical techniques are a new paradigm in Veterinary patients also as is the case in humans. Laparocsopy assisted surgery in veterinary practice is getting popularized in teaching institutes and slowly percolating into field practice also. The cost of the equipment is quite prohibitive for its routine use in veterinary practice. Innovative approaches to devise novel techniques , instruments and methods to reduce costs, surgery time and excessive manipulation of viscera are being pursued of late in Veterinary practices. This work is a nice attempt on a very important clinical condition presented in horses. The manuscript is nicely written and well illustrated with figures and photographs. The limitation of the study is very less number of cases and that too lead to failure of devised equipment during the surgery leading to delayed surgery time. A surgical instrument giving way during surgery puts a serious question mark on its efficacy. I appreciate that the authors candidly reported this limitation and have acknowledged this limitation of their innovation. It is nice attempt on which future work can be built upon.

Specific comments

Line 44: Please replace ‘discoveries’ with ‘innovations’

Line 65: Please replace ‘discoveries’ with ‘innovations’

Lines 68-69: Please reframe as ‘However, no report was seen by the authors on the use of a low-cost multiport device for single access upon perusal of literature’

Line 70-72: Please reframe as ‘Therefore, the objective of this study was to develop and evaluate a new low-cost multiport device for performing laparoscopy assisted cryptorchidectomy in standing horses’

Line 77: Please delete ‘guided’

Lines 79-80: Please replace ‘home- made’ with ‘self innovated’

Line 93: Please replace ‘home- made’ with ‘self innovated’

Line 149: Please reframe as ‘The intra abdominal inspection’

Lines 164-67: Please reframe as ‘The evaluation of the new single-port multiple-access self innovated device in live horses was carried out. Four cryptorchid purebred horses, over three years of age of the Manga- larga, Lusitano, quarter horse and crossbreed breeds presented at the veterinary hospital for routine laparoscopic cryptorchidectomy formed the subjects of this study’

Line 168: Please delete ‘, non-simultaneously and’

Line 170: Please reframe ‘After a total fast of 12 hours’ as ‘After 12hours fasting’

Line 174: What is meant by ‘associated’? Please clarify.

Lines 316-18: Earlier it was mentioned that the thickness of the abdominal wall did not allow the port to be inserted and here it is mentioned that the adhesions did not allow the removal of the testicle. Please clarify and correct accordingly.

Line 347: Please replace ‘associated with’ with ‘comprising of’

Line 393: What is meant by ‘sperm structures’? Please clarify.

Line 395: How can it be claimed as safer than normal laparoscopic techniques? Please clarify.

Lines 419-22: Using multiple ports in laparoscopy may not induce more trauma as manipulation through single port can take more time and induce more trauma. Does this device reduce the manipulation and ultimately the surgery time? Please clarify.

Lines 446-58: This device was handmade single port device. Please check that you are at times advocating a multiport device leading to am ambiguity to the study. Please clarify and correct.

Comments on the Quality of English Language

Quality of english language used is fine with few grammatical errors. At few places the sentences are too long and ambiguous.

Author Response

(The authors gave the same response as above.)

Round 2

Reviewer 2 Report

Comments and Suggestions for Authors

It just occurred to me after second review that the title is misleading as it claims a single-port access while the multiport device is combined with a second port to insert a laparoscope in the abdomen.

I suggest to review this title to represent the reality.

The whole Introduction needs grammatical revision for too long sentences and construction.

You describe the device for standing laparoscopy but its design doesn't show any contraindication to use in horses under general anaesthesia. This aspect has no reason to be emphasized.

Upon which evidence do you base your statement of lines 58-59?

Line 75 "self-innovated" do you mean self-invented?

All instruments are by pronouns and need a capital letter ex: Babcock

Line 157 and below: is it relevant to indicate the suture pattern of the flank incisions of your cadavers in the ex-vivo component of the study?

line 163: at the start of the sentence you mention purebred but the last animal is a crossbreed one. Breed are pronouns and need a Capital letter

line 240, can you consider that 500 ml of Hartmann constitutes an intraabdominal lavage or did you infuse this solution with 2% Lidocaine for pain management purpose. 2ml of 2% Lidocaine remains a very small dose for any of the patients described

line 245: dermorrhaphy not dermorrhaphia, why not simply write skin suture?

line 271, availability of material and cost are not mentioned in your method or data set collection, you need to add them in your methodology or remove them from you result and eventually mention them in the discussion

line 281 - 285: does your custom made device play any role in the capacity to perform a wide exploration of the abdomen, it is simply what has aleady been described elsewhere by placing a laparoscope in the paralumbar fossa. This paragraph doesn't present any innovation related to your device

line 292: this sentence should be in your method and you have already mentioned this at line 262, at this section you should only mention "the table below presents the times..."

line 301 sedation and analgesia are not parameters of your material and methods

line 307, your project doesn't assess the quality of exploration of the abdomen by a laparoscope placed in the paralumbar fossa

line 320 adhesiolysis - was it with more then 1 hand?

line 336 how can a device of 5.5 cm diameter fit in an incision of 2.5 cm?

line 367: this sentence is obsolete, that data is known since decades

line 374 this sentence is incomprehensible

line 375 this segment would be justified if your device was developed to prevent general anaesthesia while all you demonstrate is that the device can be used in the standing restrained horse

line 389 sentence is incomplete

line 415: what is the meaning of this sentence?

line 416: what is the meaning of this sentence?

line 417 greater safety then ? as this project isn't involving a control of any kind, this superiority has no basis

line 434 complete and comprehensive exploration of the abdominal cavity - isn't it necessary to procise that it is only of the ipsilateral half abdomen???

Comments on the Quality of English Language

English requires extensive improvement

Author Response

Comments and Suggestions for Authors

It just occurred to me after second review that the title is misleading as it claims a single-port access while the multiport device is combined with a second port to insert a laparoscope in the abdomen.

I suggest to review this title to represent the reality.

Response: Title reformulated “Single-port multiple-access custom made device used in laparoscopic assisted cryptorchidectomy in standing horses - a pre-liminary study”

The whole Introduction needs grammatical revision for too long sentences and construction.

Response: Introduction reformulated in the manuscript

            Laparoscopic cryptorchidectomy is one of the most commonly performed procedures in the field of equine genitourinary surgery, despite limitations related to the surgeon skill and use of specialized equipment [1,2,3]. Cryptorchidectomy assisted with laparoscopy can allow better intraabdominal inspection, facilitating access and manipulation of the testicle retained in the cavity [1,4, 5].

In recent decades, there have been many advances related to laparoscopic cryptorchidectomy in horses, the main benefits of which are the absence of general anesthesia and better visualization of the cryptorchid [1,6,7]. Standing horse laparoscopy is the best option for removing ectopic testicles, but complications can occur when the devices are not used during access to the abdominal cavity [8,9].

Related advances to laparoscopic surgery depend on improvements and related innovations to the technique and equipment [1,9,10]. The most recent studies demonstrate single access through an umbilical port for removal of cryptorchid testicles and the use of devices such as wound retractor and resorbable self-locking loop [7,11, 12].

No report was seen by the authors on the use of a low-cost multiport device for laparoscopic assisted cryptorchidectomy in horses. The objective of this study was to develop and evaluate a new low-cost multiport device for performing laparoscopy-assisted cryptorchidectomy in horses. Our hypothesis is that it would be possible to develop a new low-cost multiport device to perform laparoscopically assisted cryptorchidectomy in horses.

You describe the device for standing laparoscopy but its design doesn't show any contraindication to use in horses under general anaesthesia. This aspect has no reason to be emphasized.

Response: The introduction has been reformulated and we believe that this emphasis has been removed.

Upon which evidence do you base your statement of lines 58-59?

Response: The evidence is based on the articles cited [1,4,5]

Line 75 "self-innovated" do you mean self-invented?

Response: Suggestion from one of the reviewers from the previous round. We believe he meant to refer to innovation.

All instruments are by pronouns and need a capital letter ex: Babcock

Response: Corrected in manuscript.

Line 157 and below: is it relevant to indicate the suture pattern of the flank incisions of your cadavers in the ex-vivo component of the study?

Response: The suture pattern was written in the manuscript.

198-202: After exploration intra-abdominal with a new device, it was removed and the pneumoperitoneum was undone, followed by myorrhaphy with separate X suture using ny-lon-0 thread.  Dermorrhaphy was performed with suture in a simple isolated pattern using nylon-0 thread.

line 163: at the start of the sentence you mention purebred but the last animal is a crossbreed one. Breed are pronouns and need a Capital letter

Response: Corrected in manuscript.

Four cryptorchid horses, over three years of age of the Manga-larga, Lusitano, quarter horse, and crossbreed Breeds presented at the veterinary hospital for routine laparoscopic cryptorchidectomy, were the subjects of this study.

line 240, can you consider that 500 ml of Hartmann constitutes an intraabdominal lavage or did you infuse this solution with 2% Lidocaine for pain management purpose. 2ml of 2% Lidocaine remains a very small dose for any of the patients described

Response: The washing performed was not for pain control; lidocaine was used in this case as an anti-adhesion solution for the tissues.

line 245: dermorrhaphy not dermorrhaphia, why not simply write skin suture?

Response: Correction made to the manuscript to “skin suture”

line 271, availability of material and cost are not mentioned in your method or data set collection, you need to add them in your methodology or remove them from you result and eventually mention them in the discussion

Response: Taken from results.

line 281 - 285: does your custom made device play any role in the capacity to perform a wide exploration of the abdomen, it is simply what has aleady been described elsewhere by placing a laparoscope in the paralumbar fossa. This paragraph doesn't present any innovation related to your device

Response: Paragraph deleted from the manuscript.

line 292: this sentence should be in your method and you have already mentioned this at line 262, at this section you should only mention "the table below presents the times..."

Response: Suggestion included in the manuscript.

line 301 sedation and analgesia are not parameters of your material and methods

Response: Sentence deleted from the manuscript.

line 307, your project doesn't assess the quality of exploration of the abdomen by a laparoscope placed in the paralumbar fossa

Response: Paragraph deleted in the manuscript.

line 320 adhesiolysis - was it with more then 1 hand?

Response: With one hand. Corrected in manuscript.

line 336 how can a device of 5.5 cm diameter fit in an incision of 2.5 cm?

Response: The 5.5 cm base is not introduced into the abdominal cavity, only the distal ring of the device. The ring is flexible and can be inserted into the abdominal cavity.

line 367: this sentence is obsolete, that data is known since decades

Response: Please describe the phrase, it is not possible to identify it with the manuscript that was recommended for download.

line 374 this sentence is incomprehensible

Response: The paragraph was removed as it discusses anesthesia during the procedure. Our work only demonstrates that the device can be used on the standing horse.

line 375 this segment would be justified if your device was developed to prevent general anaesthesia while all you demonstrate is that the device can be used in the standing restrained horse

Response: The paragraph was removed as it discusses anesthesia during the procedure. Our work only demonstrates that the device can be used on the standing horse.

line 389 sentence is incomplete

Response: The sentence was reformulated in the manuscript, we hope to have resolved this understanding.

line 415: what is the meaning of this sentence?

Response: The paragraph was removed as it discusses anesthesia during the procedure. Our work only demonstrates that the device can be used on the standing horse.

line 416: what is the meaning of this sentence?

Response: The paragraph was removed as it discusses anesthesia during the procedure. Our work only demonstrates that the device can be used on the standing horse.

line 417 greater safety then ? as this project isn't involving a control of any kind, this superiority has no basis

Response: The paragraph was removed as it discusses anesthesia during the procedure. Our work only demonstrates that the device can be used on the standing horse.

line 434 complete and comprehensive exploration of the abdominal cavity - isn't it necessary to procise that it is only of the ipsilateral half abdomen???

Response: Suggestion included in the manuscript.

Reviewer 3 Report

Comments and Suggestions for Authors

Dear authors !

Thank you for submitting the revised version of the manuscript. I see improvements in several instances, but I see that you did not respond to several of my questions and concerns (highlighted in yellow in the attached document).

Please have a look on these.

Author Response

Dear authors !

Thank you for submitting the revised version of the manuscript. I see improvements in several instances, but I see that you did not respond to several of my questions and concerns (highlighted in yellow in the attached document).

From MM on several questions were not answered:

Figure 1: I’ve suggested to unpack the tracheal tube to improve understanding of the construct and to improve the quality of the photo as it is not very clear.

Response: Figure 1 was redone and included in the manuscript, as suggested. We hope we have answered.

Additional comment

2.4. How can you investigate feasibility of cryptrochidectomy in bovine fetuses (did they have abdominal testicles?), otherwise you should only say that you investigate feasibility of using your device and introducing instruments. Why did you use bovine fetuses and not horses euthanized as this would be more representative of what you wish to test?

Response: Sacrificed horses were not used because our group did not have one at the time, however, we believe that tests on cadavers in these experimental surgeries are important to test new techniques and use of new instruments.

Correction made to the manuscript “The evaluation of the multiple access device in bovine fetal cadavers was to verify the feasibility of using this device and introducing instruments into the abdominal cavity. Bovine fetuses were used because they are constantly used in our routine investigations and discoveries of video-surgical techniques.”

Figure 2: Thank you for adding the additional photo of intrabdominal view.

Photo B not clear how the custom-made device is placed in the flank?

Response: The sentence was added in the caption, we hope we have fulfilled the suggestion “… , distal ring introduced into the abdominal cavity through minilaparotomy, …”

2.5. Did the horses present unilateral cryptorchidism or bilateral? How did you manage the normally descended testicle?

Response: Unilateral. The testicle was pulled and manipulated with the Babcock forceps inserted by the device.

Evaluation in cryptorchid horses

The device does not seem sufficiently elaborated for publication. Of the 4 horses, you could not even use it in one. Furthermore, as you state that you could not remove the testicle in 2 horses via laparoscopy, as one horse had an adhesion, and one other horse had a testicle that was in the inguinal canal. One of these cases was the case where you could not enter your custom-made device (as the abdominal wall was too thick).

Response: Yes. One was the one in which we were unable to insert the device and the other was the horse that had the testicle in the inguinal canal strongly adhered.

The surgery time in cryptorchid horses was very long for removal of only one testicle (or did you do more than this ?).

Illustration of the removed testicles should have a scale for the size.

Response: Size scale inserted in the image and the figure inserted in the manuscript.

The testicles seem very small and therefore the small incision size is not only linked to your device but also to the small size of the testicles. Please include this into the discussion.

Response: Sentence included in the discussion (374-376)

“The surgical scar was reduced as it was a minimally invasive procedure, and the small size of the testicle removed was another important factor in determining the size of this scar.”

The short surgery time in bovine foetuses should not be overemphasised as in these cases you did not do anything except inspection and you do not face bleeding like in live animals.

In the following lines are those of the marked manuscript annexed to the cover letter.

Line 459-463 When you talk about “inguinal region”, you mean the abdomen near to the inguinal ring? What do you mean by “ectopic location”?

Response: Yes. We refer to the region close to the inguinal ring.

As for ectopic location, we refer to the testicle outside its normal location (scrotum).

Line 491-492 What do you mean by this? In general surgical time was reduced in patients? Your surgical time was very long.

And the whole paragraph 491-500?

Response: Sentence excluded from the manuscript.

"In general, surgical time was reduced in patients, as they were animals of different sizes with testicles retained in different locations."
